# Tensor Implementation of Monte-Carlo Tree Search for Model-Based Reinforcement Learning

**Marek Baláž** * and **Peter Tarábek**

Faculty of Management Science and Informatics, University of Žilina, Univerzitná 8215/1, 010 26 Žilina, Slovakia
* Correspondence: marek.balaz@fri.uniza.sk

**Abstract:** Monte-Carlo tree search (MCTS) is a widely used heuristic search algorithm. In model-based reinforcement learning, MCTS is often utilized to improve action selection process. However, model-based reinforcement learning methods need to process large number of observations during the training. If MCTS is involved, it is necessary to run one instance of MCTS for each observation in every iteration of training. Therefore, there is a need for efficient method to process multiple instances of MCTS. We propose a MCTS implementation that can process batch of observations in fully parallel fashion on a single GPU using tensor operations. We demonstrate efficiency of the proposed approach on a MuZero reinforcement learning algorithm. Empirical results have shown that our method outperforms other approaches and scale well with increasing number of observations and simulations.

**Keywords:** Monte-Carlo tree search; reinforcement learning; MuZero; parallel computations; tensor GPU implementation; model-based reinforcement learning

## 1. Introduction

Reinforcement learning (RL) is a core machine learning topic that is concerned with how agents should perform actions in an environment in order to maximize the total cumulative reward. The agent learns by interacting with the environment through trial and error and uses feedback from its own actions and experiences. Its purpose is to find optimal or nearly-optimal strategy that maximizes the reward function. This strategy is referred to as policy. RL based methods has achieved outstanding accomplishments in a number of domains, e.g., games [1,2], autonomous driving [3], UAVs [4–6], robotics [7], and traffic signal control [8].

RL may be divided into two fundamental categories: model-free, and model-based. Model-free approaches directly learns a value function or a policy by interacting with the environment. Model-based RL uses model of environment in order to perform decision making through planning. This model is commonly represented by a Markov decision process (MDP) [9] consisting of two components: a state transition function, and a reward function. MDP is widely used in artificial intelligence for modeling sequential decision-making scenarios with probabilistic dynamics [10–12].

One of the most employed planning techniques is Monte-Carlo tree search (MCTS). MCTS combines the precision of tree search with the generality of random sampling [13]. It is a well-established approach to look-ahead search in context of model-based RL to extend and improve decision making process by building and traversing a tree. It was utilized in AlphaGO [1], the first program to achieve superhuman performance in GO, and in many of its successors [2,14,15].

MCTS is very challenging to parallelize due to its inherent sequential nature where each rollout depends on the statistics computed from previous simulations [16]. This problem is even more challenging when GPU architecture is involved. One of the problems is the SIMD execution scheme within GPU which causes that a standard CPU parallel implementation such as root-parallelism fail [17].

In this paper, we propose a parallel implementation that aims to evaluate multiple unique MCTS trees and is fully implemented on a graphics processing unit (GPU). A common way to use MCTS is to gradually build one large tree. This tree is continuously used and updated for the needs of the task at hand. We focus on a different type of tasks, namely tasks that need to evaluate a large number of unique trees instead of building one tree gradually. We show that the GPU tensor implementation of MCTS is suitable for this task and can outperform CPU and CPU-GPU implementations despite the atomic nature of the operations in MCTS. As an example of such task, we present a model-based RL, which was our prime motivation in developing this implementation. Here, a large number of different states (observations) need to be evaluated in each training iteration. Each state is represented as a root in a unique tree, which is then processed by MCTS. These trees are built in order to obtain actual training data and are then discarded and rebuilt from new states in a subsequent training iteration. Efficient generation and evaluation of new trees during the training is the key to the performance of these methods. Therefore, our implementation can dramatically speed up the training process through more efficient evaluation of large number of observations.

To fully exploit GPU's capabilities, an environment model should also be implemented on the GPU. This is crucially important for overall GPU implementation of MCTS. Model-based RL can be divided into methods with an explicitly given model and with a learned model. For methods with an explicitly given model, the possibility of implementing the model on GPU as well as its complexity is highly application dependent. In the case of methods with a learned model, the feasibility and efficiency of the implementation depends on the employed learning algorithm to learn model dynamics.

Inspired by the recent success of MuZero [2], we demonstrate efficiency of the proposed MCTS implementation on a MuZero algorithm. MuZero is a model-based RL method that learns dynamics model within its search. We show that the proposed implementation can be easily integrated with a learned dynamics model represented by deep neural network (DNN). Although in this paper we focus on a parallel implementation of MCTS in conjuction with MuZero, our implementation is broadly applicable to all model-based RL methods that utilized MCTS. The contribution of this work is the following:

- We propose fully parallel GPU implementation of MCTS that can simultaneously evaluate MCTS on large number of observations. We used number of observations 50 - 750 for our evaluation and show that the proposed method scales well with the increasing number of observations. The code is available at https://github.com/marrekb/MuZero (accessed on 11 December 2022).
- We compare our method with existing MCTS implementations on a use case inspired by the MuZero and Atari games domain. The choice of DNN architecture used in RL agent is application dependent and have a significant impact on the overall performance. Therefore, we also report results for setup without using DNN. A pattern can be seen in the results which shows that the proposed method is the most computationally efficient.

The rest of the paper is organized as follows. The upcoming Section 2 contains an overview of related work. We discuss important role of MCTS in RL methods on example of MuZero, and existing parallel approaches to MCTS. In Section 3 we describe our proposed parallel implementation of MCTS. Section 4 is devoted to experiments and their evaluation. We summarize our conclusions in Section 5.

## 2. Related Work

### 2.1. Mcts and Reinforcement Learning

RL agents are usually divided into two categories—model-based, and model-free. One of the most important parts of model-based agents is the model of the environment. By environment model, we mean a function that predicts the next state and reward given a combination of the current state and action. In other words, the model allows inferences to be made about how the environment will behave [18]. Therefore, these models are

frequently used for planning, e.g., in combination with MCTS. Agents such as AlphaGO and AlphaZero use a given model of the environment in the form of game simulators to achieve the best results. Other model-based agents such as MuZero, SimPLE [19], and I2A [20] do not require a given model and instead incorporate a learned model into the training procedure.

Model-free agents do not use a model of the environment, so they cannot reason about how their environments will change based on a given state and action [18]. Model-free agents rely on a trial-and-error approach. They tend to be easier to implement and tune. Model-free agents are divided by their learning approach into Q-learning agents (e.g., DQN [21]) and policy optimization agents (e.g., A2C [22], PPO [23]).

In 2016, the algorithm AlphaGO defeated world champion Lee Sedol in game GO [1]. This event was another milestone in the dominance of artificial intelligence against human players. Last generation of algorithms based on AlphaGO is MuZero. It is model-based RL algorithm that combines MCTS and DNN [2]. It is useful in domain without own copy of environment, e.g., domain of Atari games. Due to this approach, MuZero has achieved SOTA results in several Atari games [24]. Atari games are often used as benchmarks for RL algorithms [21,25]. In addition of Atari games, MuZero has been applied in real-world tasks such as autonomous defense of SDN Networks [26] or air traffic optimization [27].

The DNN in MuZero consists of three connected components for representation, dynamics and prediction. The representation function $f_r(o|\theta_r)$ takes past observation (e.g., the Go board or Atari screen) and transforms it into hidden state $s$. The hidden state $s$ is a representation of the real environmental state by DNN. Because we do not have a copy of environment, we use hidden states in MCTS instead of environment observations.

Given a state $s_t$ and an action $a_t$, the dynamics function $f_d(s, a|\theta_d)$ predicts an immediate reward $R(s_t|\theta_d)$ and a new state $s_{t+1}$. In each state, the prediction function $f_p(s|\theta_p)$ proposes probability distribution of actions $P(s, \cdot|\theta_p)$ and state value $V(s|\theta_p)$. State value is expected return of rewards from the current state.

MCTS is used to build tree of possible actions (represented as edges) and hidden states (nodes) in order to support decision making process. Tree is build by process called simulation. Simulation consists of four base MCTS phases (selection, expansion, simulation and backpropagation). Each simulation explores existing tree, finds next unexplored combination of state $s$ and action $a$, and extends the tree by adding new node (state $s$). Simulations are executed sequentially. One simulation consists of following phases:

- Phase of selection—the tree is traversed from the root to find next unexplored combination of state and action. In each node (assigned to specific state), the edges (represent actions) are chosen by the PUCT (Predictor + Upper Confidence Bound applied to trees) (Equation (1)). PUCT represents trade-off between exploitation and exploration. Exploitation is solved via q values. $Q(s_t, a)$ is expected value of future rewards after taking action $a$ [18]. In our case, it could be computed as a sum of immediate reward $r_t$ and state value of next state $V(s_{t+1}|\theta_p)$ discounted by discount factor $\gamma \in [0, 1]$ (Equation (2)). In board games, rewards usually mean results of games (1 for a win, 0 for a draw and $-1$ for a loss). In domain of Atari games, rewards are obtained either during or at the end of the game. Rest of Equation (1) forms the exploration part. $P(s_t, a|\theta_p)$ represents predicted probability of action $a$ by the prediction function, $c_1$ and $c_2$ are exploration constants.
- Phase of expansion—After selection of unexplored combination of state $s_t$ and $a_t$, new node (with state $s_{t+1}$) is added to the tree structure. State $s_{t+1}$ is obtained by using dynamic function with selected combination $s_t$ and $a_t$ as an input.
- Phase of simulation—during the phase of simulation, DNN is called again, specifically prediction function to obtain probability distribution (used in PUCT equation) and state value (used to compute q values). These variables are assigned to new node.
- Phase of backpropagation—q value computed from state value is back propagated through trajectory composed of each traversed node during the phase of selection. Each traversed node's number of visits $N(s_t, a_t)$ is increased by one.

$$a_t = \underset{a}{\mathrm{argmax}} \, Q(s_t, a) + P(s_t, a | \theta_p) \cdot \frac{\sqrt{\sum_b N(s_t, b)}}{1 + N(s_t, a)}$$
$$\cdot \left( c_1 + \log \left( \frac{\sum_b N(s_t, b) + c_2 + 1}{c_2} \right) \right) \tag{1}$$

$$Q(s_t, a_t) = \mathbb{E} \left[ r_t + \gamma \cdot V(s_{t+1} | \theta_p) \right] \tag{2}$$

Unlike the original MCTS, MuZero creates new tree of specific size in each step of the game to avoid a cumulative error of predicted states. The size of the tree depends on the number of simulations $C_S$ (e.g., 800 simulations per state in board games or 50 simulations in the domain of Atari games [2]).

Each MCTS node stores following values:

- State $s$
- Predicted probability distribution of actions $P(s, \cdot | \theta_p)$
- Vector $N(s, \cdot)$ representing the number of visits of each action
- Vector $Q(s, \cdot)$ of q values for each action
- Vector $R(s, \cdot)$ of rewards for each action
- Set of children nodes

In the theory, most of RL papers explain MCTS as a single tree. However, in practice it is appropriate to play many games in multiple processes in order to collect data as soon as possible. As we mentioned before, in case of MuZero, new tree is build in each step of game. Therefore, multiple trees are build at the same time in many processes separately. Also each process uses DNN with the same weights.

### 2.2. Parallel Approaches to MCTS

In the last two decades, different approaches have been proposed to obtain parallel implementation of MCTS phases. Parallelization via CPU is useful, if high performance shared-memory machines are available [28]. According to [29], we can divide the basic methods into leaf parallelization, root parallelization and tree parallelization. Leaf parallelization methods are easiest to implement. Each time a node is expanded, multiple parallel simulations (playouts) are performed. After all playouts are completed, the combined score is propagated to improve the accuracy of the node value. In root parallelization, multiple search trees are constructed by separate threads and combined together occasionally. The most frequently used methods to combine the values from different trees are average voting and majority voting [30]. These methods tend to have the least communication overhead. Tree parallelization uses multiple threads to update a single search tree at different nodes. Since multiple threads can update the same node, data corruption can occur.

Steinmetz and Gini [30] compared the benefits of root parallelization to tree parallelization and measured both against a baseline of building a larger tree utilizing more time. They obtained the results in the Go domain on CPU hardware and show that parallel algorithms keep pace with or may exceed the performance gained by increasing the amount of time. Soejima et al. [31] analyzed the performance of two root parallelization strategies: average voting and majority voting. Their results with 64 CPU cores and computer Go programs showed that majority voting outperforms average voting. Rocki and Suda [17] proposed a hybrid CPU-GPU parallel MCTS based on the Block-parallel scheme. The method runs several trees in root parallel fashion on the CPU and blocks of leaf parallel playouts on the GPU. In this approach, the GPU kernel is called asynchronously and the control is given back to CPU. Barriga et al. [32] proposed Multiblock Parallel algorithm. The method is similar to the [17], but instead of running simulations only for one child of the selected node, they run simulations for all children. The goal is to take full utilization of GPU. Świechowski and Mańdziuk [33] investigate the concept of parallelization of MCTS on general game playing framework. They proposed Limited Hybrid Root-Tree Parallelization method. The idea is to combine root and tree parallelization approaches. They use tree parallelization within one CPU machine which is then mixed by root parallelization between

the machines. Liu et al. [34] proposed parallelization of MCTS with UCT modification - Balance Unobserved in UCT (BU-UCT). BU-UCT is designed to efficiently divide nodes during the selection phase into multiple process workers. These workers are responsible for implementing the next three phases.

The mentioned approaches focus on parallelization of operations in one huge MCTS tree. However, our goal is to implement fully GPU parallelization which focuses on processing a large number of relatively small MCTS trees instead of single large one. The proposed implementation deals directly with parallelization of multiple MCTS trees in conditions with one or small number of GPUs. Implementations that address the parallelization of multiple MCTS trees are based on fully CPU or semi-GPU parallelization. The closest to our work are Werner [35] and EfficientZero [36] implementations.

Werner's open-source implementation of MuZero uses multiprocessing during data collection. In each process, the copy of DNN is held. Only one game runs per process, so the parallelization of MCTS is designed as one MCTS per process. The implementation uses Ray library utilizing a single GPU per process. Therefore, in computers with a lower number of GPUs it is better to use fully CPU implementation. Werner's implementation has been widely used [26,27,37,38] and there are many other implementations based on the same principle available on GitHub [39–44].

EfficientZero is more sample efficient version of original MuZero. In their implementation, each process also holds current copy of the DNN on GPU. Each process collects data from multiple games. MCTS method is applied to all observations simultaneously, so that multiple trees are built in method (one tree per game). During MCTS simulation, phase of selection is executed sequentially for each tree. Selected states and actions are sent as batch into DNN. New nodes are sequentially created based on the obtained data (new states with probability distributions and state values). During backpropagation phase, nodes of traversed trajectories are also updated sequentially.

## 3. Proposed Implementation

In the case of Werner implementation, the inefficient parts are computation of prediction for each tree separately and higher communication overhead (e.g., sending collected data to the main process, and updating the DNNs in child processes). In the case of the EfficientZero, the disadvantage is the sequential processing of most MCTS phases in each processed tree.

Our implementation is based on tensor operations because they can be automatically parallelized on the GPU. Therefore, in each phase all trees are processed in parallel. We used Python library for deep learning—PyTorch [45]. However, the proposed method can be easily implemented in other libraries such as TensorFlow [46].

### 3.1. Data Structure and Notation

In this section we introduce data structure and the established notations. Our data structure consists of the multiple tensors to store different attributes of MCTS nodes.

During the design, we were inspired by the structure of the Q table. Q table is a matrix holding q values. Rows represent states and columns possible actions in environment. Although Q table stores all possible states, we need to store significantly smaller number of states. MCTS method is applied on batch which consists of $C_T$ observations (i.e., running environments) with $C_S$ MCTS simulations per observation. Therefore, we need to store $C_T + C_S \times C_T$ states. First $C_T$ states are roots obtained from observations via representation function. Next $C_S \times C_T$ states will be explored and stored during the MCTS simulations ($C_T$ states per simulation). In our implementation we have to leave the zero row empty for implementation reasons (explained later), so the total number of rows in each tensor is $C_R = C_T + C_S \times C_T + 1$.

Data structure of proposed method (shown in Figure 1) consists of following tensors:

- Tensor $S$—all states are held in tensor $S$. The number of dimensions of tensor $S$ is equal to the number of dimensions of the state $+1$. The index of the first dimension

represents unique IDs that are shared across all tensors. States are added into tensor $S$ during the initialization (roots) and simulations (explored states) in order that they were visited (roots have indices from 1 to $C_T$, explored states from the first simulation have indices between $C_T + 1$ and $2 \times C_T$ and so on).

- Tensor $Q$—q values are stored in tensor $Q$, similar to the Q table. First index (of the row) is the same unique ID of the node as we mentioned in tensor $S$. The size of tensor is $C_R \times |A|$. At the beginning of MCTS method, all values in tensor Q are initialized to zeros. During the phase of backpropagation, q values of traversed nodes are updated.
- Tensor $R$—holds the predicted rewards by dynamics function. It works on the same principle as tensor $Q$ and has the same size $C_R \times |A|$. Rewards are put into tensor during phase of expansion.
- Tensor $P$—stores the predicted probabilities computed by prediction function. The size is similar like previous two tensors $C_R \times |A|$.
- Tensor $N$—unlike the previous tensors composed by real numbers, tensor $N$ consists of integers. It stores the numbers of visits of nodes (executed combinations of states and actions). The size of tensor is again $C_R \times |A|$.
- Tensor $E$—the last tensor $E$ (with the size $C_R \times |A|$) holds IDs of children nodes. For example, if there is an edge from the parent node $ID = i$ after taking action $a$ to the children node $ID = j$, then $E(i, a) = j$. If there is no edge between two nodes, the value in tensor is zero. For that reason, zeroth row is empty in each tensor.

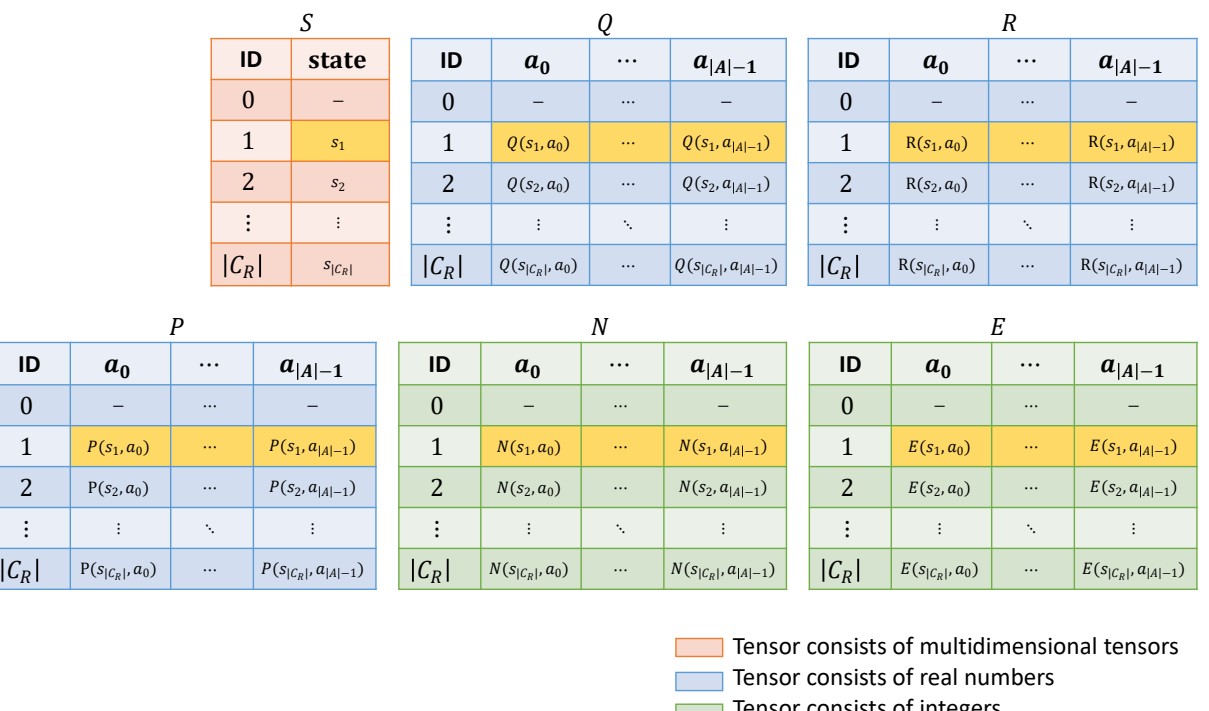

**Figure 1.** Data structure—the data of each node is stored in 6 tensors. IDs represent indices of nodes (they are not stored as tensor values). For example, the highlighted rows represent all stored data of a node with ID = 1.

The complete data structure consists of the above mentioned 6 tensors. The first index of each tensor is the ID of the node (index of the row). In the tensor $S$, index $i$ returns $i$-th state (of the node with $ID = i$) whereas in other tensors $i$-th vector is returned instead. E.g., in tensor $Q$, the index $i$ returns vector of q values $Q(i) = [Q(i, 0), Q(i, 1), \cdots, Q(i, |A| - 1)]$. On the other hand, index $i$ in combination with action index $a$ returns scalar q value $Q(i, a)$.

In our implementation, we often index via vectors (as a part of tensor operations). E.g., combination of node index $i$ and vector of actions $\vec{a} = [a_0, a_{15}, a_{17}]$, where $a_0, a_{15}$ and $a_{17}$ represent integer numbers, applied to tensor $Q$ results in $Q(i, \vec{a}) = [Q(i, a_0), Q(i, a_{15}), Q(i, a_{17})]$.

Indexing by combination of two vectors $\vec{i} = [i_2, i_5, i_1, i_{25}]$ and $\vec{a} = [a_{10}, a_5, a_{17}, a_{10}]$ applied to tensor $Q$ lead to $Q(\vec{i}, \vec{a}) = [Q(i_2, a_{10}), Q(i_5, a_5), Q(i_1, a_{17}), Q(i_{25}, a_{10})]$.

### 3.2. Initialization and Preprocessing

At the beginning of the method, local variables are initialized (Algorithm 1, lines 1–9) based on the obtained parameters $o$, $C_S$, and $|A|$. Vector of unique root indices $\vec{i}_R = [1, 2, 3, \cdots, C_T - 1, C_T]$ is generated. These indices are utilized as unique identifiers across all tensors.

The tensor of root states $s$ is computed from the obtained batch of observations by representation function. The tensor of probabilities and vector of values are predicted by the prediction function. Dirichlet noise is added to the tensor of probabilities in order to support the exploration of MuZero. States and tensors of probabilities are inserted into tensors $S$ and $P$ via a vector of root indices (Algorithm 1, lines 14–15).

---

**Algorithm 1** Initialization

---

**Require:** batch of observations $o$
**Require:** the number of MCTS simulations $C_S$
**Require:** the number of actions $|A|$
1: $C_T \leftarrow |o|$
2: $C_N \leftarrow C_S + 1$
3: $C_R \leftarrow 1 + C_T \times C_N$
4: $S \leftarrow$ tensor of zeros with size $C_R \times state\ dim$
5: $P \leftarrow$ tensor of zeros with size $C_R \times |A|$
6: $Q \leftarrow$ tensor of zeros with size $C_R \times |A|$
7: $R \leftarrow$ tensor of zeros with size $C_R \times |A|$
8: $N \leftarrow$ integer tensor of zeros with size $C_R \times |A|$
9: $E \leftarrow$ integer tensor of zeros with size $C_R \times |A|$
10: $\vec{i}_R \leftarrow$ integers from the interval $[1, C_T]$
11: $s \leftarrow f_r(o\,|\theta_r)$
12: $p, \vec{v} \leftarrow f_p(s\,|\theta_p)$
13: $p \leftarrow p +$ Dirichlet noise
14: $S(\vec{i}_R) \leftarrow s$
15: $P(\vec{i}_R) \leftarrow p$

---

### 3.3. Phase of Selection

The pseudocode of the phase of selection is given in Algorithm 2, and the flowchart is shown in Figure 2. During one iteration of MCTS, each tree is traversed to find the leaf node. Let the $L$-step trajectory consists of combinations of traversed nodes and edges (actions) in order, then we can write trajectory as $\tau = [(ID_0, a_0), (ID_1, a_1), (ID_2, a_2), \cdots, (ID_{L-1}, a_{L-1})]$. Tensor $I$ holds IDs of traversed nodes and tensor $A$ indices of selected actions chosen by the PUCT method. Both tensors have size $C_N \times C_T$. $C_N$ is the number of simulations increased by one because the first ID and action in trajectory belong to the root of a tree. Each tree has its trajectory stored in one column whose index corresponds to a particular tree. Therefore, the number of columns is $C_T$. Vector $\vec{l}$ stores the indices of the last elements of each trajectory. Information stored in all three tensors are used in the next phases.

Each trajectory starts from the root. Root indices (IDs) are inserted into the zero row of tensor $I$ (Algorithm 2, line 4).

Indices of active trajectories are stored in vector $\vec{i}_N$. In other words, vector $\vec{i}_N$ remembers indices of trees that are still active in the phase of selection. At the beginning of each selection phase, vector is filled by vector of all tree indices because each tree takes a part in the selection phase.

The loop of the selection phase starts with condition $\vec{i}_N \neq \varnothing$. Condition checks whether the vector of active trajectories is empty. If there is no active trajectory, the phase of selection is finished. Otherwise, IDs of nodes in active trajectories are assigned to vector $\vec{i}$ by

indexing via the current step and vector $\vec{i_N}$ in the tensor $I$ (Algorithm 2, line 8). Computing of the PUCT method (Equation (1)) for current nodes is fully executed by tensor operations of adding, multiplying, dividing, etc. The method returns vector of selected actions of current nodes.

Indices of the last items in active trajectories are updated (Algorithm 2, line 11).

The IDs of the nodes accessed using the actions $\vec{a}$ in current nodes $\vec{i}$ are obtained from tensor $E$. As we mentioned before, if there is no children node for combination $((ID, a))$, 0 is returned instead. Obtained IDs and possible zero indices are inserted into a new row of the tensor $I$.

For simplicity's sake, let the number of trees $C_T = 4$, number of active trajectories $|\vec{i_N}| = 3$, indices of active trajectories $\vec{i_N} = [0, 1, 3]$, node IDs $\vec{i} = [5, 7, 8]$, selected actions $\vec{a} = [3, 1, 0]$ and IDs of children nodes $E(\vec{i}, \vec{a}) = [E(5, 3) = 15, E(7, 1) = 0, E(8, 0) = 21] = [15, 0, 21]$. Since $I(step, \vec{i_N}) = [15, 0, 21]$, updated row of the tensor is $I(step) = [15, 0, 0, 21]$. $I(step, 0) = 15$ and $I(step, 3) = 21$ indicate active trajectories. $I(step, 1) = 0$ represents inactive trajectory. This trajectory has been terminated in the current step because the node with $ID = 7$ has no children, e.g., $E(7, 1) = 0$. $I(step, 2) = 0$ represents a trajectory that was already inactive.

Finally, we update the indices of nonzero values from the current row of $I$. If there is no nonzero value, the phase of selection is completed. On the other hand, if there is at least one nonzero value (active trajectory), the next phase of the MCTS simulation begins.

---

**Algorithm 2** Phase of selection

---

1: $I \leftarrow$ integer tensor of zeros with size $C_N \times C_T$
2: $A \leftarrow$ integer tensor of zeros with size $C_N \times C_T$
3: $\vec{l} \leftarrow$ integer vector of zeros with length $C_T$
4: $I(0) \leftarrow \vec{i_R}$
5: $step \leftarrow 0$
6: $\vec{i_N} \leftarrow$ integers from the interval $[0, C_T)$
7: **while** $\vec{i_N} \neq \emptyset$ **do**
8:    $\vec{i} \leftarrow I(step, \vec{i_N})$
9:    $\vec{a} \leftarrow$ apply PUCT on nodes $\vec{i}$
10:    $A(step, \vec{i_N}) \leftarrow \vec{a}$
11:    $\vec{l}(\vec{i_N}) \leftarrow step$
12:    $step \leftarrow step + 1$
13:    $I(step, \vec{i_N}) \leftarrow E(\vec{i}, \vec{a})$
14:    $\vec{i_N} \leftarrow$ return indices of nonzero values from $I(step)$
15: **end while**

---

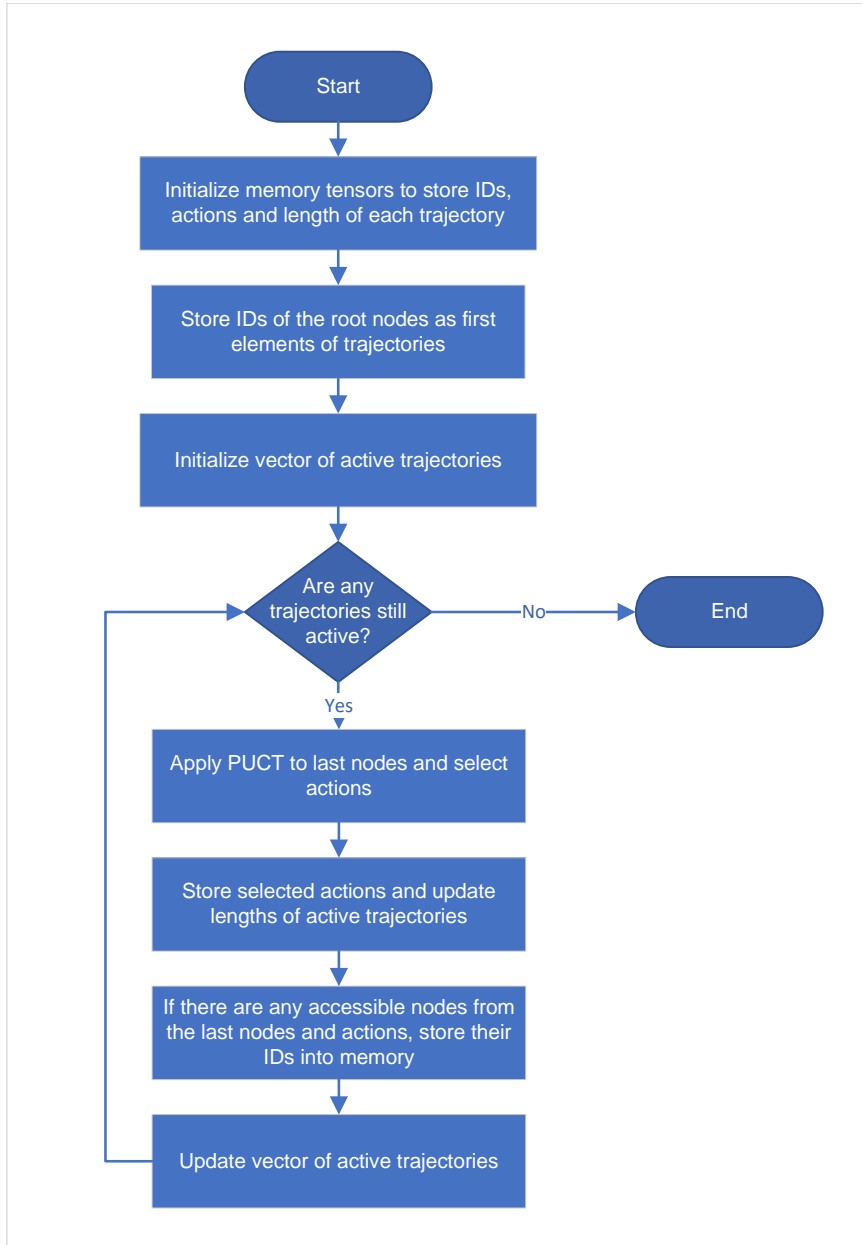

**Figure 2.** Flowchart of the phase of selection.

*3.4. Phase of Expansion and Simulation*

We joined phases of expansion and simulation (Algorithm 3 and Figure 3) because both phases are interconnected and it is effective to implement them together. Based on the vector of last indices $\vec{l}$, combinations of last nodes and actions are identified for all trajectories (Algorithm 3, lines 1–3). Tensor of new states $s$ and vector of obtained rewards $\vec{r}$ are computed by dynamics function. The new states are then used in the prediction function to predict tensor of probabilities $p$ and vector of values $\vec{v}$.

IDs of new nodes are computed from the vector of roots $\vec{i_R}$. Tensors of children nodes $E$ and rewards $R$ are updated by new data.

At the end of the expansion phase, predicted tensors of states $s$ and probability distributions $p$ are added to the tensors $S$ and $P$ as data of new nodes (Algorithm 3, lines 9–10).

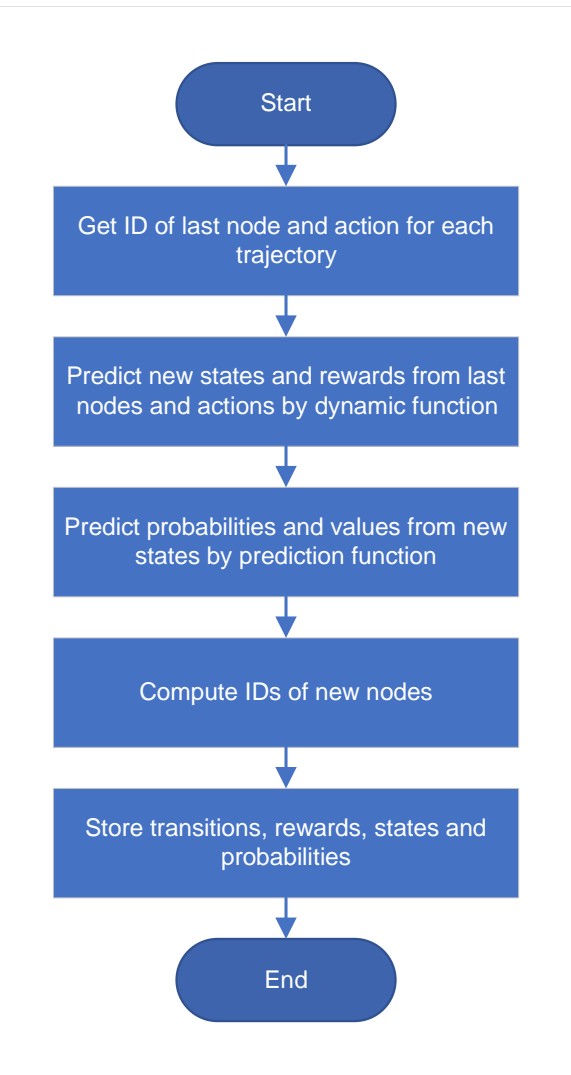

**Figure 3.** Flowchart of the phase of expansion and simulation.

---

**Algorithm 3** Phase of expansion and simulation

---

**Require:** current number of simulation $C_I$

1:  $\vec{k} \leftarrow$ integers from the interval $[0, C_T - 1]$
2:  $\vec{i} \leftarrow I(\vec{l}, \vec{k})$
3:  $\vec{a} \leftarrow A(\vec{l}, \vec{k})$
4:  $s, \vec{r} \leftarrow f_d(S(\vec{i}), \vec{a}|\theta_d)$
5:  $p, \vec{v} \leftarrow f_p(s|\theta_p)$
6:  $\vec{i_{new}} \leftarrow \vec{i_R} + C_T \times (C_I + 1)$
7:  $E(\vec{i}, \vec{a}) \leftarrow \vec{i_{new}}$
8:  $R(\vec{i}, \vec{a}) \leftarrow \vec{r}$
9:  $S(\vec{i_{new}}) \leftarrow s$
10:  $P(\vec{i_{new}}) \leftarrow p$

---

### 3.5. Phase of Backpropagation

During the phase of backpropagation (Algorithm 4 and Figure 4), trajectories are traversed from the end to the start to update q values and numbers of visits. At the beginning of the backpropagation phase, the local variable *step* stores the length of the longest trajectory obtained in the selection phase. Therefore, the loop of backpropagation starts from $step - 1$.

We first select the trajectory indices that were active in the selection phase during the actual *step*. Based on the obtained indices, IDs and actions in actual *step* are identified from tensors *I* and *A* (Algorithm 4, lines 3–4).

Values of active trajectories $\vec{v}(\vec{i_N})$ are used to update q values of parent nodes (values of non-active trajectories remain unchanged). In the end, tensors *Q* and *N* are updated by standard formulas (Equation (4) of [2]).

The loop of backpropagation ends by updating q values and the number of visits of root nodes.

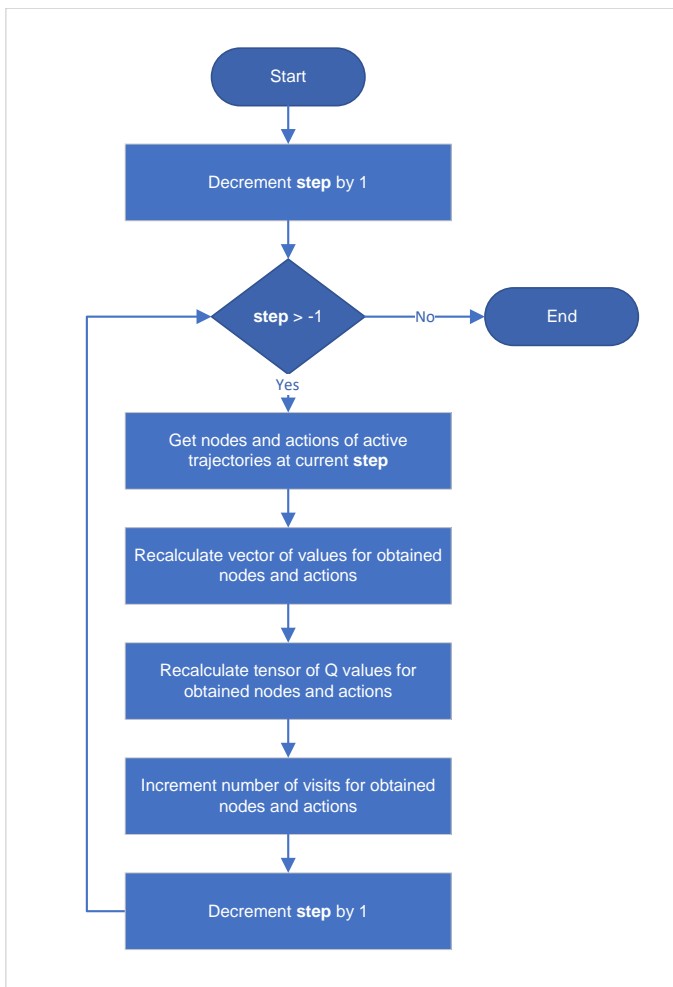

**Figure 4.** Flowchart of the phase of backpropagation.

---

**Algorithm 4** Phase of backpropagation

1: **for** $step \leftarrow$ decrement from $step - 1$ to 0 **do**
2: $\quad \vec{i_N} \leftarrow$ return indices of nonzero values from $I(step)$
3: $\quad \vec{i} \leftarrow I(step, \vec{i_N})$
4: $\quad \vec{a} \leftarrow A(step, \vec{i_N})$
5: $\quad \vec{v}(\vec{i_N}) \leftarrow R(\vec{i}, \vec{a}) + \gamma \times \vec{v}(\vec{i_N})$
6: $\quad Q(\vec{i}, \vec{a}) \leftarrow \frac{Q(\vec{i},\vec{a}) \times N(\vec{i},\vec{a}) + \vec{v}(\vec{i_N})}{N(\vec{i},\vec{a})+1}$
7: $\quad N(\vec{i}, \vec{a}) \leftarrow N(\vec{i}, \vec{a}) + 1$
8: **end for**

---

*3.6. Post Processing*

The MCTS method completes after $C_S$ simulations have been executed. At the end of the MCTS method, it is necessary to calculate the probability distribution and the state value of each root state. Probability of root action $a$ is computed as $\frac{N(s_{root}, a)}{\sum_b N(s_{root}, b)}$. The values of the root states are computed as a weighted arithmetic mean of root q values and their probability distributions. All computations are carried out on GPU exploiting tensor operations for adding, multiplication and division.

## 4. Experiments

The proposed implementation aims at the evaluation of a large number of MCTS instances in parallel. We demonstrate this capability with an example from the RL domain where a large number of observations need to be processed at the same time. Each observation represents the state in the environment (root state in MCTS tree) in which we want to perform an action. MCTS algorithm is utilized in RL to improve action selection process.

We compared proposed implementation to three MCTS implementations used in model-based RL methods. We measured the performance of individual implementations in terms of how long it takes to simultaneously evaluate MCTS on batch of observations. For each experiment, we report the mean and standard deviation calculated from 100 replications. All experiments were carried out on a single NVIDIA GeForce RTX 2080 Ti graphics card and 16-core Intel(R) Xeon(R) CPU E5-2643 @ 3.30GHz processor.

The list of compared implementations together with their designation is as follows:

- GPU—The proposed method fully implemented on GPU using tensor operations and described in Section 3.
- CPUGPU_P—Implementation inspired by EfficientZero [36] which uses multiple processes. Each process builds multiple trees and stores copy of DNN on GPU. However, maintaining a copy of DNN by children processes is memory inefficient which limits their use especially in single GPU scenarios. Therefore, in our modification, only the parent process stores a DNN on GPU.
  At the beginning of MCTS method, batch of observations is given to representation and prediction functions (on GPU) to predict states, probability distributions and state values in roots. Predicted data are split and sent to the children processes in which trees are initialized (one tree per root's state). MCTS simulations are executed in each process. Phase of selection is executed sequentially on CPU. States of selected nodes and actions are sent back into parent process. After receiving data from all children processes, new states, rewards, probabilities and values are predicted by one forward of dynamics and prediction functions (on GPU). Predicted data are split and sent again into children processes in which phases of expansion and backpropagation are performed (also on CPU). After executing MCTS simulations, the parent process receives and post-processes the results.
- CPUGPU_S—A sequential implementation of CPUGPU_P method without multiprocessing. Phases of selection, expansion and backpropagation are performed on CPU inorder. As in the previous implementation, all data is processed as a batch by DNN on GPU. Both CPUGPU_P and CPUGPU_S were implemented by our team and used as a part of AlphaZero in [47].
- CPU—Last approach represents Werner's MCTS implementation. We used the source code from author's GitHub repository [35].

*4.1. Model of Environment*

We chose MuZero algorithm as an example of model-based RL method that utilize MCTS. Our experimental setup was inspired by the domain of Atari games. One observation (state of the environment) was represented by $128 \times 96 \times 96$ tensor. The number of possible actions was set to 18, which is the maximum amount of possible actions in domain

of Atari games. MuZero algorithm uses DNN to approximate representation, dynamics and prediction function. We use original MuZero's architecture [2] with a few modifications.

The kernel size is $3 \times 3$ for all operations. The convolution operations padding is set to 1. Representation function is identical copy of MuZero's original function. It consists of:

- 1 convolution with stride 2 and 128 kernels, output resolution $48 \times 48$
- 2 residual blocks with 128 kernels
- 1 convolution with stride 2 and 256 kernels, output resolution $24 \times 24$
- 3 residual blocks with 256 kernels
- average pooling with stride 2, output resolution $12 \times 12$
- 3 residual blocks with 256 kernels
- average pooling with stride 2, output resolution $6 \times 6$

Output size of representation function, i.e., state size is $256 \times 6 \times 6$.

The input to the dynamics function is $257 \times 6 \times 6$ dimensional tensor consisted of state and action. Action represents tensor $1 \times 6 \times 6$ filled by value $\frac{a_t}{count\ of\ actions}$. The dynamics function provides two outputs. The first is the new state and the second is the reward. The reward in original MuZero implementation was computed as a linear combination of categorical output. We formulated reward prediction problem as a regression task instead. The second change from the original implementation is in the structure of hidden layers of reward head as they were not described in the documentation.

Layers of dynamics functions are:

- 1 convolution with 256 kernels, output resolution $6 \times 6$
- 8 residual blocks with 256 kernels, output resolution $6 \times 6$ (the last residual block is also used as a representation of the new state)
- flattening, number of output neurons is 9216
- fully connected layer with 512 neurons
- fully connected layer with 1 neuron (reward output)

Outputs of representation and dynamics functions are used as inputs for prediction function. It also provides two outputs—probability distribution and state value. The common part is composed of flattening and linear layer with 512 neurons. Each output head consists of one linear layer with 512 neurons. Last linear layer of probability distribution has 18 neurons. As in the case of the reward head, last output layer of the state value head has one neuron. Again, the categorical task has been reformulated into a regression task.

Our goal was not to train MuZero agent but to compare the computational speed of individual MCTS implementations. Therefore, we used a randomly initialized DNN model to simulate all necessary computations.

### 4.2. MCTS Parameters

Batch of observations is generated as a random tensor of size $C_T$. Each of these observations is processed by a unique instance of MCTS with the number of simulations set to $C_S$.

Although MCTS uses PUCT formula to select action, we modified the action selection mechanism to test the edge cases of tree formation based on the following scenarios:

1. Random action—in this scenario, the action is selected randomly. This scenario causes the tree to build in breadth (tree resembles a balanced tree). The scenario reflects, for example, the behavior at the beginning of RL agent training, when the DNN produces approximately uniform probability distribution of actions.
2. Constant action—in this scenario, the selected action is replaced by a constant action. This scenario causes the tree to build in depth (tree resembles a linked list). The scenario reflects, for example, the behavior of the trained or overfitted agent, when the DNN produces one dominant action.

The use of scenarios is implemented by changing the value of probability distribution function according to the given scenario and setting the scalar values (e.g., state value and reward) to zero.

### 4.3. Results

We report results for two sets of experiments. In the first set, we measured the performance of MCTS adjusted for the effect of DNN model used by MuZero. Architecture of DNN utilized by MuZero is strongly dependent on the application. Therefore we first provide performance of compared implementations without the DNN model. Second set of experiments shows results on the use case of MuZero with DNN model described earlier.

Results for experiments without DNN model are shown in Tables 1 and 2. We fixed the number of simulation $C_S$ to 50. Batch of observations size $C_T$ was set to 50, 100, 250, 500 and 750. In the case of CPUGPU_P implementation, we report results for number of processes set to 2, 5, 10 and 25. For CPU implementation, we report results with number of processes set to 5, 10 and 15. In both cases, a larger and smaller number of processes resulted in higher computation time.

**Table 1.** Effect of number of observations on computation time. Results of experiments for random action without DNN model ($C_S$ = 50).

| $C_T$ | GPU | CPUGPU_P | | | | CPUGPU_S | CPU | | |
|---|---|---|---|---|---|---|---|---|---|
| | | **2** | **5** | **10** | **25** | | **5** | **10** | **15** |
| 50 | 0.225 ±0.001 | 1.721 ±0.013 | 2.045 ±0.01 | 2.891 ±0.017 | 5.576 ±0.038 | 4.09 ±0.003 | 7.62 ±0.112 | 15.147 ±0.223 | 22.999 ±0.183 |
| 100 | 0.234 ±0.001 | 2.855 ±0.009 | 2.832 ±0.011 | 3.749 ±0.016 | 6.282 ±0.032 | 8.001 ±0.005 | 20.849 ±0.036 | 33.206 ±0.139 | 44.956 ±0.635 |
| 250 | 0.243 ±0.001 | 6.27 ±0.016 | 4.869 ±0.022 | 5.103 ±0.014 | 7.224 ±0.035 | 19.865 ±0.008 | 68.015 ±0.158 | 90.801 ±0.427 | 116.284 ±0.755 |
| 500 | 0.253 ±0.001 | 11.421 ±0.022 | 7.447 ±0.029 | 7.299 ±0.026 | 8.811 ±0.031 | 39.61 ±0.015 | 174.611 ±0.361 | 193.741 ±0.239 | 248.905 ±0.738 |
| 750 | 0.262 ±0.001 | 16.131 ±0.045 | 9.791 ±0.041 | 9.118 ±0.022 | 10.368 ±0.038 | 60.344 ±0.015 | 303.172 ±0.536 | 323.785 ±1.458 | 396.275 ±0.724 |

**Table 2.** Effect of number of observations on computation time. Results of experiments for constant action without DNN model ($C_S$ = 50).

| $C_T$ | GPU | CPUGPU_P | | | | CPUGPU_S | CPU | | |
|---|---|---|---|---|---|---|---|---|---|
| | | **2** | **5** | **10** | **25** | | **5** | **10** | **15** |
| 50 | 1.686 ±0.008 | 7.519 ±0.012 | 5.194 ±0.028 | 5.095 ±0.019 | 7.096 ±0.028 | 35.175 ±0.064 | 14.941 ±0.119 | 14.49 ±0.143 | 17.266 ±0.127 |
| 100 | 1.704 ±0.006 | 13.575 ±0.012 | 7.825 ±0.03 | 7.24 ±0.018 | 8.812 ±0.033 | 69.282 ±0.116 | 28.769 ±0.044 | 28.142 ±0.035 | 47.295 ±0.12 |
| 250 | 1.732 ±0.005 | 31.876 ±0.027 | 15.257 ±0.031 | 13.037 ±0.028 | 12.975 ±0.033 | 170.135 ±0.163 | 72.939 ±0.067 | 72.173 ±0.152 | 147.788 ±0.395 |
| 500 | 1.766 ±0.004 | 63.233 ±0.041 | 28.529 ±0.033 | 22.916 ±0.044 | 20.886 ±0.037 | 340.716 ±0.474 | 144.271 ±0.205 | 140.942 ±0.088 | 374.888 ±0.753 |
| 750 | 1.787 ±0.003 | 94.143 ±0.087 | 41.317 ±0.042 | 31.937 ±0.031 | 28.512 ±0.046 | 500.181 ±0.479 | 218.951 ±0.395 | 207.96 ±0.137 | 685.497 ±1.611 |

In both random action and constant action scenarios we see a similar pattern. Proposed GPU implementations is the fastest for all tested values of the $C_T$ parameter. The difference in speed increases with increasing number of observations. In Figure 5 we show

detail comparison between two best performing implementations for both scenarios. For $C_T = 750$, proposed GPU implementations is 34.8 times faster for random action scenario and 16 times faster for constant action scenario than the second best performing CPUGPU_P implementations (with 10 and 25 processes respectively). In the constant action scenario (Table 2), the computational time increased for most of the investigated implementations and their settings. This increase is due to a growing trajectories obtained by the phase of selection in MCTS.

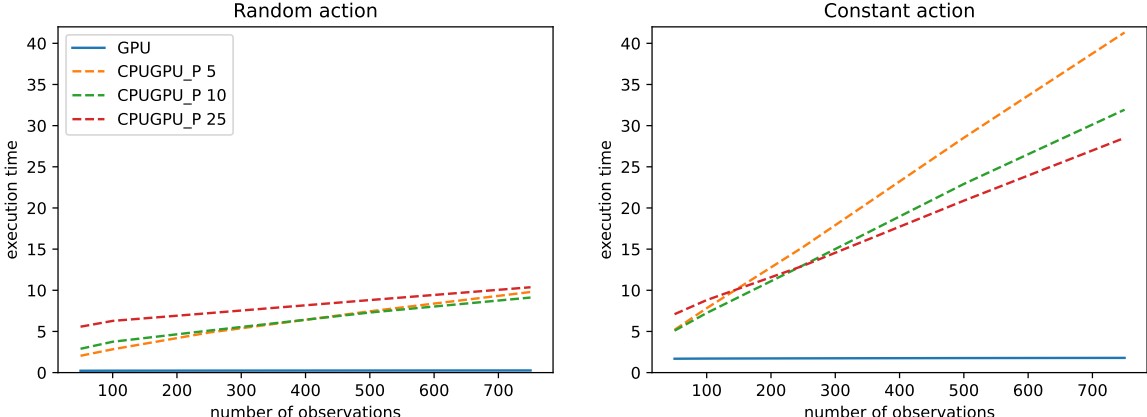

**Figure 5.** Comparison of execution time of the two best performing implementations with respect to the number of observations. Results for both scenarios without DNN.

We further investigated influence of the number of MCTS simulations $C_S$ and report results for two best performing implementations in Tables 3, 4 and Figure 6. The $C_T$ parameter was set to 100. We can observe that the computation time increases significantly as the number of simulations increases, especially in constant action scenario. Proposed GPU implementation is 12.1 times faster for $C_S = 50$ and 8.7 times faster for $C_S = 400$ in random action scenario. In constant action scenario, GPU implementation is 4.2 times faster for $C_S = 50$ and 2 times faster for $C_S = 400$.

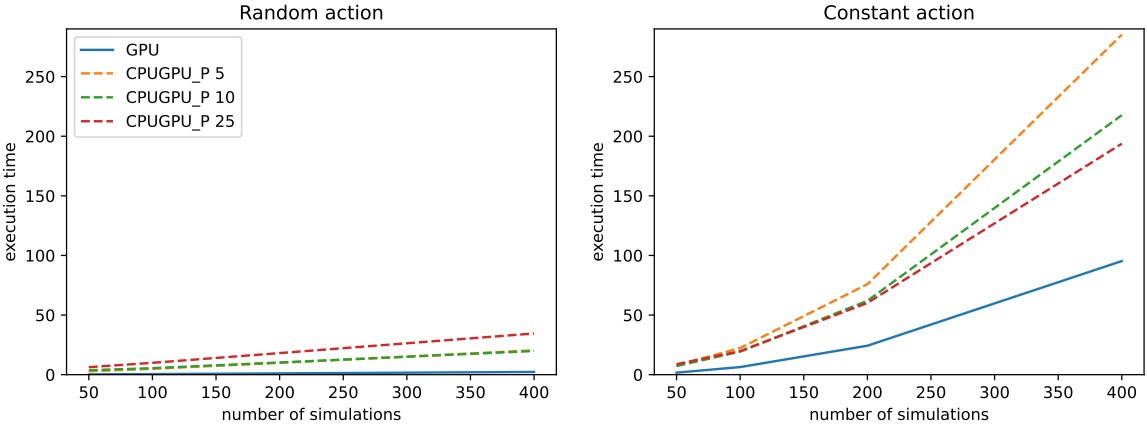

**Figure 6.** Comparison of execution time of the two best performing implementations with respect to the number of simulations. Results for both scenarios without DNN.

**Table 3.** Effect of number of simulations on computation time. Results of experiments for random action without DNN ($C_T = 100$).

| $C_S$ | GPU | CPUGPU_P | | |
|---|---|---|---|---|
| | | 5 | 10 | 25 |
| 50 | 0.234 ±0.001 | 2.832 ±0.011 | 3.749 ±0.016 | 6.282 ±0.032 |
| 100 | 0.481 ±0.006 | 4.906 ±0.015 | 5.454 ±0.019 | 9.945 ±0.032 |
| 200 | 1.068 ±0.001 | 9.908 ±0.028 | 10.225 ±0.023 | 18.109 ±0.034 |
| 400 | 2.294 ±0.002 | 20.422 ±0.134 | 19.844 ±0.031 | 34.562 ±0.082 |

**Table 4.** Effect of number of simulations on computation time. Results of experiments for constant action without DNN ($C_T = 100$).

| $C_S$ | GPU | CPUGPU_P | | |
|---|---|---|---|---|
| | | 5 | 10 | 25 |
| 50 | 1.704 ±0.006 | 7.825 ±0.03 | 7.24 ±0.018 | 8.812 ±0.033 |
| 100 | 6.371 ±0.029 | 22.36 ±0.072 | 19.344 ±0.03 | 19.734 ±0.037 |
| 200 | 24.281 ±0.135 | 75.867 ±0.095 | 61.968 ±0.066 | 60.144 ±0.099 |
| 400 | 95.291 ±0.571 | 285.212 ±0.216 | 217.657 ±0.139 | 193.77 ±0.1 |

Last experiments measured performance with DNN model. We omitted CPU method from results due to the high computation requirements associated with the execution of DNN model (e.g., 338 s for $C_S = 50$, and $C_T = 50$). Similarly to previous experiments, we fixed $C_S$ to 50 and set $C_T$ to 50, 100, 250, 500 and 750. Results presented in Tables 5, 6 and Figure 7 show similar trend as in experiments without DNN. Proposed GPU is the best performing implementation for all values of parameter $C_T$. Even with the DNN model, which consumes most of the computation, GPU implementation is 4.7 times faster for random action scenario and 7.7 times faster for constant action scenario than the second best performing CPUGPU_P implementations (with 10 and 25 processes respectively), for $C_T = 750$.

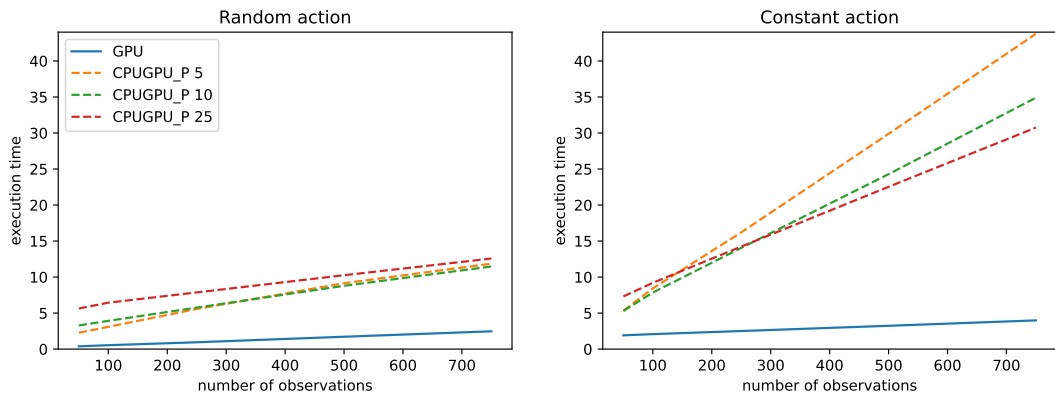

**Figure 7.** Comparison of execution time of the two best performing implementations with respect to the number of observations. Results for both scenarios with DNN.

**Table 5.** Effect of number of observations on computation time. Results of experiments for random action with DNN model ($C_S$ = 50).

| $C_T$ | GPU | CPUGPU_P | | | | CPUGPU_S |
| | | 2 | 5 | 10 | 25 | |
|---|---|---|---|---|---|---|
| 50 | 0.383 ±0.001 | 1.962 ±0.007 | 2.269 ±0.008 | 3.285 ±0.016 | 5.634 ±0.047 | 6.091 ±0.004 |
| 100 | 0.539 ±0.001 | 3.053 ±0.006 | 3.091 ±0.008 | 3.919 ±0.015 | 6.442 ±0.037 | 10.001 ±0.005 |
| 250 | 0.946 ±0.001 | 6.909 ±0.012 | 5.574 ±0.014 | 5.757 ±0.021 | 7.868 ±0.026 | 21.864 ±0.009 |
| 500 | 1.714 ±0.002 | 12.607 ±0.019 | 9.144 ±0.044 | 8.787 ±0.023 | 10.254 ±0.03 | 41.61 ±0.018 |
| 750 | 2.466 ±0.003 | 18.185 ±0.024 | 11.857 ±0.055 | 11.47 ±0.039 | 12.571 ±0.031 | 62.345 ±0.016 |

**Table 6.** Effect of number of observations on computation time. Results of experiments for constant action with DNN model ($C_S$ = 50).

| $C_T$ | GPU | CPUGPU_P | | | | CPUGPU_S |
| | | 2 | 5 | 10 | 25 | |
|---|---|---|---|---|---|---|
| 50 | 1.909 ±0.008 | 7.942 ±0.011 | 5.332 ±0.023 | 5.296 ±0.019 | 7.309 ±0.028 | 33.371 ±0.05 |
| 100 | 2.074 ±0.007 | 14.182 ±0.022 | 8.426 ±0.028 | 7.849 ±0.023 | 9.201 ±0.026 | 67.45 ±0.105 |
| 250 | 2.507 ±0.005 | 32.975 ±0.028 | 16.207 ±0.028 | 14.054 ±0.035 | 14.231 ±0.033 | 168.301 ±0.142 |
| 500 | 3.232 ±0.004 | 65.08 ±0.053 | 29.884 ±0.035 | 24.283 ±0.035 | 22.527 ±0.048 | 338.874 ±0.423 |
| 750 | 3.984 ±0.004 | 97.403 ±0.092 | 43.776 ±0.041 | 34.896 ±0.04 | 30.752 ±0.046 | 501.425 ±0.351 |

## 5. Conclusions

In this paper, we proposed a parallel implementation of MCTS that efficiently evaluates large number of MCTS trees at once. It utilizes tensor operations and is fully implemented on GPU. We show that the atomic nature of MCTS operations can be transformed into vector operations suitable for GPU. We demonstrated this capability using the example of MuZero model-based RL agent in Atari game domain. Model-based RL agents often combine DNN and MCTS approaches to improve action selection. During the offline training, these RL agents requires to process a huge amount of observations in parallel. These observations are represented by unique root nodes in MCTS.

We compared our implementation with approaches based on the Werner and EfficientZero implementations. We show that the proposed approach gives the best results and scales well with the number of observations and number of simulations. We tested two scenarios of tree formations: random action and constant action. For both scenarios the proposed implementations yield the best results. In experiments without DNN, the proposed implementation is 34.8 times faster for the random action and 16 times faster for the constant action scenario than the second best performing CPUGPU_P implementation ($C_T$ = 750 and $C_S$ = 50).

We further investigated the effect of DNN in RL agent. In experiments with DNN and for the random action, the proposed implementation is 4.7 times faster than the second best performing CPUGPU_P for the value of parameters $C_T$ = 750 and $C_S$ = 50. In the case

of the constant action, this difference is 7.7 fold. Therefore, we observed a performance improvement over the benchmark implementations in both non-DNN and DNN settings.

Although we report results when MCTS is utilized within the MuZero RL agent, the proposed implementation can be used wherever a large number of MCTS instances need to be processed in parallel. The closest example to us is the use in the model-based RL. We show an example of model-based MuZero agent which uses DNN to learn model dynamics. The proposed implementation can also be utilized in model-based RL agent with an explicitly given model of environment. In this case the overall performance depends on the complexity of environment and its implementation.

Our implementation was tested using a single GPU. We see no restrictions for deployment on a larger number of GPUs. However, for more complex computing infrastructures, we expect that directly tailored methods will yield better results, as they can exploit the specifics of a given infrastructure.

**Supplementary Materials:** The following supporting information can be downloaded at: https://www.mdpi.com/article/10.3390/app13031406/s1.

**Author Contributions:** Conceptualization, M.B. and P.T.; methodology, M.B. and P.T.; software, M.B.; validation, M.B. and P.T.; formal analysis, M.B. and P.T.; investigation, M.B.; resources, M.B. and P.T.; data curation, M.B.; writing—original draft preparation, M.B. and P.T.; writing—review and editing, M.B. and P.T.; visualization, M.B. and P.T.; supervision, P.T.; project administration, M.B. and P.T.; funding acquisition, P.T. All authors have read and agreed to the published version of the manuscript.

**Funding:** This publication was realized with support of Operational Program Integrated Infrastructure 2014–2020 of the project: Intelligent operating and processing systems for UAVs, code ITMS 313011V422, co-financed by the European Regional Development Fund.

**Institutional Review Board Statement:** Not applicable.

**Informed Consent Statement:** Not applicable.

**Data Availability Statement:** Source codes and raw experiment results are available on https://github.com/marrekb/MuZero (accessed on 11 December 2022) in the folder "proposed_mcts". Data presented in this study are available in Supplementary Materials.

**Conflicts of Interest:** The authors declare no conflict of interest.

## Abbreviations

The following abbreviations are used in this manuscript:

| | |
|---|---|
| MCTS | Monte-Carlo tree search |
| RL | reinforcement learning |
| MDP | Markov decision process |
| GPU | graphics processing unit |
| CPU | central processing unit |
| DNN | deep neural network |

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
