# Peer review of "Tensor Implementation of Monte-Carlo Tree Search for Model-Based Reinforcement Learning"

_applsci, doi:10.3390/app13031406_

Round 1
Reviewer 1 Report
Dear Authors,
Your work addresses an important and pressing topic in RL. Below you will find my comment section by section.
1. Introduction
Please mention references for categories of RL. Additionally, please add further explanation of model-based RL and model-free.
Similarly, for MDP as well add suitable references.
It's a journal paper, so avoid "popular" or "commonly." However, "widely used " could be used with suitable references.
Adding research questions or summarizing it would make the article more readable.
In line 71, could the author give a reference in % how much better their approach is? Can it happen that for some other data set, the result would be different ??--> The proposed method achieved the best execution time for both setups.
2. Related work
It needs to be longer and needs to add further literature. Additionally, adding a bit of taxonomy of RL will help the readers and future researchers. Something along the line of (https://link.springer.com/article/10.1007/s10796-022-10314-0).
3. Proposed Implementation
I like the section. Very well written and explained. However, adding graphics showing the flow of implementation and approach would be highly recommended.
4. results
great work, well-written section.
5. Conclusion
After reading the paper, I understood the practical contribution of the work completed. However, I could not see any significant theoretical contribution to the literature. The authors need to Highlight how their work contributes to the literature on RL.
Reviewer 2 Report
The paper propose a parallel implementation of MCTS using a GPU and tensor operations. The related works and the state of the art is well represented and give a solid overview in order to understand how the proposed MCTS implementation go beyond the current state-of-the-art. The experimental part is well structured and demonstrate in a clear way how the proposed parallel approach is able to lower the computational time required by MCTS in a RL scenario, using as a reference the existing implementations strategies described in the related works section.
The overall quality of the paper can be improved with the following minor modification:
- In section 3, line 176, it can be useful to clarify in a more extensive way the meaning of "higher maintenance associate with multiprocessing" in the contest of the MCTS Werner implementation.
- The data structure described in section 3.1 is very complex and a graphical representation will help the reader and the overall description.
- The phases described in sections from 3.1 to 3.6 can be represented on a flow-diagram.
- The example in section 3.3 can be coupled with a graphical representation that will help the reader and give a clear description of the selection process.
- In section 4.3 it can be useful to give an explanation related to the kernel size (even if empirical).
Finally, the github repository will be more suitable for possible implementations if well documented (the code structure and quality is good, but the lack of comments/documentation within the code doesn't facilitate the usage by other developers/scientist).
